# Palmitoylation of the Bovine Foamy Virus Envelope Glycoprotein Is Required for Viral Replication

**DOI:** 10.3390/v13010031

**Published:** 2020-12-27

**Authors:** Keli Chai, Zhaohuan Wang, Yali Xu, Junshi Zhang, Juan Tan, Wentao Qiao

**Affiliations:** Key Laboratory of Molecular Microbiology and Technology, Ministry of Education, College of Life Sciences, Nankai University, Tianjin 300071, China; 1120170375@mail.nankai.edu.cn (K.C.); 2120171034@mail.nankai.edu.cn (Z.W.); 2120181068@mail.nankai.edu.cn (Y.X.); 1120180399@mail.nankai.edu.cn (J.Z.); juantan@nankai.edu.cn (J.T.)

**Keywords:** bovine foamy virus, envelope glycoprotein, palmitoylation, BDHHC3, BDHHC20, membrane fusion, subviral particle, cell surface, replication

## Abstract

Membrane proteins of enveloped viruses have been reported to undergo palmitoylation, a post-translational modification often having a critical role in the function of these viral proteins and hence viral replication. In this study, we report that the foamy virus (FV) envelope (Env) glycoprotein is palmitoylated. Specifically, we found that bovine foamy virus (BFV) Env (BEnv) is palmitoylated at amino acid positions C58 and C59 by BDHHC3 and BDHHC20 in a DHHC motif-dependent manner. In addition, mutations C58S and C58/59S significantly decrease cell surface expression of BEnv, subviral particle (SVP) egress, and its membrane fusion activity, thus ultimately inhibiting BFV replication. The C59S mutation exerts a minor effect in this regard. Taken together, these data demonstrate that the function of BEnv in the context of BFV replication is under the regulation of palmitoylation.

## 1. Introduction

Foamy viruses (FVs, also termed spumaviruses) constitute the only genus of the Spumaretrovirinae subfamily in the Retroviridae family for their unique replication strategy [1,2]. FVs are nonpathogenic retroviruses that cause endemic infection in non-human primates, felines, equines, and bovines [3]. They can be transmitted to humans through zoonotic infection [4,5,6]. As a member of the understudied non-human Spumaretrovirinae subfamily, bovine foamy virus (BFV) is highly prevalent in bovines without inducing overt diseases [7,8]. However, BFV is detectable in the general human food chain through beef and raw milk, which poses a potential risk of zoonotic transmission of BFV to humans [8]. Owning to their non-pathogenic nature and broad tissue tropism, FVs have been engineered as safe viral vectors for gene transfer as well as gene therapy [9,10].

Similar to other retroviruses, the typical viral genome of FVs encodes three canonical structural proteins: group specific antigen (Gag), polymerase (Pol), and envelope (Env) glycoprotein [1,11]. The Env protein plays an indispensable role in FV’s replication cycle not only by mediating viral entry into host cells by receptor binding and membrane fusion, but also by supporting the egress of viral particles into the cytoplasm [12,13,14]. FV entry receptor is poorly characterized so far, although it is reported that heparan sulfate might serve as a key attachment factor for prototype foamy virus (PFV) entry [11]. The host entry receptor for BFV remains largely uncharacterized. In addition, FV Env is the only viral envelope protein that specifically interacts with cognate Gag in comparison to other retroviruses, which is strictly required for targeting of FV Gag to the cellular membrane and for FV budding and particles release; thus, its function cannot be performed by heterologous viral glycoproteins [15].

FV Env is synthesized as a precursor protein and goes through a series of modifications during transportation to the cell surface before being incorporated into the budding retroviral particle [16]. FV Env is synthesized as a full-length precursor protein into the rough endoplasmic reticulum (RER) with both its N- and C-terminus located in the cytoplasm, then is cleaved by a cellular furin or furin-like protease into at least three mature subunits, an N-terminal signal or leader peptide (LP), a central surface (SU), and a C-terminal transmembrane subunit (TM) [12,15]. Subsequently, prototype foamy virus (PFV) Env undergoes two different kinds of post-translational modifications (PTMs), N-glycosylation that plays a critical role in PFV particle release and infectivity and ubiquitination that regulates subviral particle (SVP) release [17,18].

Palmitoylation (or S-palmitoylation) refers to covalent attachment of the 16-carbon saturated fatty acid to cysteine residues via a reversible thioester bond [19]. In the past few decades, a variety of viral membrane-associated proteins, especially Env proteins, have been reported to undergo palmitoylation [20,21]. Palmitoylation of viral Env proteins is crucial for regulating their trafficking, subcellular localization, translocation, and targeting to lipid rafts, membrane fusion activity, incorporation into virions, as well as viral particles assembly and release [20,21]. In human immunodeficiency virus type 1 (HIV-1), palmitoylation is vital for Env localization in lipid rafts and viral infectivity [22]. In Moloney murine leukemia virus (M-MuLV), palmitoylation of Env is critical for its lipid raft association and cell surface expression, but not required for membrane fusion activity [23]. In the case of feline immunodeficiency virus (FIV), palmitoylation of Env is essential for its fusion ability as well as incorporation into virions [24]. For Rous sarcoma virus (RSV), palmitoylation of Env is important for its plasma membrane expression and incorporation into particles [25].

Protein palmitoylation is catalyzed by palmitoyl acyltransferases (PATs), also known as DHHC proteins containing a highly conserved zinc finger Asp–His–His–Cys motif as their catalytic center [26,27]. DHHCs catalyze protein palmitoylation through a two-step reaction. The first step is auto-palmitoylation in which the DHHC protein transfers palmitate from palmitoyl-CoA to itself. Then, the enzyme transfers palmitate from itself onto the substrate protein [26,28]. In mammals, a family of 23 DHHC enzymes family has been identified [29]. DHHCs are also found in other eukaryotes. There is no evidence that any viral protein has PAT activity and function. However, viruses have evolved to hijack host PAT machinery to modify their own proteins [20,30,31,32,33,34]. For example, chikungunya virus takes advantage of DHHC2 and DHHC19 to modify its non-structural protein 1 (NSP1) [31]. Herpes simplex virus 1 (HSV-1) utilizes DHHC3 to palmitoylate its membrane protein UL20 [32,33]. HIV-1 uses DHHC20 to palmitoylate its regulatory protein Tat [34].

However, it is currently unclear whether FV Env is also palmitoylated or if palmitoylation has any role in the regulation of FV Env function and FV replication. Moreover, only few studies have investigated cellular DHHCs catalyzing acylation of viral proteins so far [30]. In this study, we report that BEnv is palmitoylated at C58 and C59 and that BDHHC3 and BDHHC20 interact with BEnv and palmitoylate BEnv at both amino acid positions. In addition, mutations of these two amino acids, particularly of C58, decrease BEnv membrane fusion ability, SVP release, and cell surface levels, thereby impairing BFV replication.

## 2. Materials and Methods

### 2.1. Plasmid Constructs

BEnv cDNA was cloned into the pCMV-3HA vector (Clonetech, Mountain View, CA, USA) using restriction enzymes EcoR I and Xho I with the HA-tag inserted at the N-terminus. Different BEnv palmitoylation site mutants (pCMV-3HA-BEnv C58S, C59S, and C58/59S) were generated using site-directed mutagenesis (Toyobo, Osaka, Japan) according to the manufacturer’s recommendations and all the mutations were verified by DNA sequencing (Genewiz, Beijing, China). The BFV infectious clones with potential palmitoylation sites mutated (pcBFV-Env C58S, C59S, and C58/59S) were generated with the pcBFV wild-type construct by replacing the EcoR I-Nde I restriction fragment corresponding to 4922–8607 bp of the wild type BFV genome with the same counterpart containing mutations.

DHHCs in the pCE-puro-3×FLAG vectors were kindly provided by Akio Kihara (Laboratory of Biochemistry, Faculty of Pharmaceutical Sciences, Hokkaido University) [35]. BDHHC3 and BDHHC20 were also inserted into the pCE-puro-3×FLAG vectors using Sal I and Not I with the FLAG-tag inserted at the N-terminus, with mutants BDHHC3-C157S and BDHHC20-C156S constructed using site-directed mutagenesis (Toyobo, Osaka, Japan) by changing Cys to Ser and verified by DNA sequencing (Genewiz, Beijing, China).

### 2.2. Cell Culture and Transfection

HEK293T, HeLa, BFVL (BHK-21-derived indicator cells containing a luciferase reporter gene driven by the BFV long terminal repeats (LTR) promoter), TZM-bl (HeLa-derived indicator cells containing a luciferase reporter gene driven by the HIV LTR promoter), and BHK-21 cells were cultured in the Dulbecco’s modified Eagle’s medium (DMEM) (Gibco, Thermo Fisher Scientific, Waltham, MA, USA) supplemented with 10% (*v*/*v*) fetal bovine serum (FBS) (Gibco), 50 U/mL penicillin, and 50 µg/mL streptomycin. All the cells were maintained at 37 °C with 5% CO_2_ in a humidified incubator.

For transfection, all cells were seeded at 70–80% confluence in either 6-, 12-well plates or 10 cm dishes. After 24 h, the cells were transfected with 1 to 4 µg of plasmids using polyethylenimine (PEI) (Polysciences, Warrington, PA, USA) with a 1:4 ratio of transfection DNA/reagent according to the manufacturer’s protocol.

### 2.3. Antibodies and Chemical Reagents

Antibodies used for protein analysis were as follows: monoclonal mouse anti-HA (catalog No. H3663, Sigma-Aldrich, St. Louis, MO, USA), monoclonal mouse anti-FLAG (catalog No. F1804, Sigma-Aldrich), polyclonal rabbit anti-Myc (catalog No. ab9106, Abcam, Cambridge, MA, USA), polyclonal rabbit anti-ATP1A1 (catalog No. 14418-1-AP, Proteintech, Chicago, IL, USA). Detection of BEnv and BGag and BTas was performed using a mouse polyclonal antibody generated in our laboratory. Alexa Fluor-488-conjugated goat anti-mouse IgG (catalog no. A-11001, Invitrogen, Carlsbad, CA, USA), HRP-conjugated goat anti-mouse IgG (catalog No. sc-2005, Santa Cruz Biotechnology, Dallas, TX, USA), and HRP-conjugated goat anti-rabbit IgG (catalog No. sc-2004, Santa Cruz).

The chemical reagents used in this paper were as follows: tris((1-benzyl-1*H*-1,2,3-triazol-4-yl)methyl)amine (TBTA, catalog No. 678937, Sigma-Aldrich), tris(2-carboxyethyl)phosphine hydrochloride (TCEP, catalog No. C4706, Sigma-Aldrich), triethanolamine (TEA, catalog No. 90279, Sigma-Aldrich), Brij^®^ O10 (catalog No. P6136, Sigma-Aldrich), CuSO_4_·5H_2_O (catalog No. C8027, Sigma-Aldrich), 2-bromopalmitate (2-BP, catalog No. 238422, Sigma-Aldrich), 4′,6-Diamidino-2′-phenylindole (DAPI, catalog No. 62248, Thermo Fisher Scientific), EDTA-free protease inhibitor cocktail (catalog No. 4693159001, Roche, Basel, Switzerland), Protein A Agarose beads (catalog No.16-125, Millipore, Boston, MA, USA). Alk-16 and az-Rho were synthesized by the Harbin Institute of Technology (Harbin, China).

### 2.4. Metabolic Labeling, Immunoprecipitation, CuAAC/Click Chemistry, and in-Gel Fluorescence Scanning

BEnv palmitoylation was performed as previously reported [36]. Transfected HEK293T cells were labeled with alk-16 (50 µM) in the DMEM supplemented with 2% FBS for 2 h at 37 °C. Labeled cells were harvested and lyzed with the pre-chilled Brij lysis buffer (1% Brij^®^ O10, 50 mM TEA, pH 7.4, 150 mM NaCl, 5×EDTA-free protease inhibitor cocktail). Cell lysates were centrifuged at 1000× *g* for 5 min at 4 °C to remove cell debris and then incubated with the mouse anti-HA antibody to immunoprecipitate BEnv proteins. For each sample, 30 µL Protein A Agarose beads were used. After 3 h incubation at 4 °C with an end-over-end rotator, the beads were washed three times with 1 mL of ice-cold radio-immunoprecipitation assay (RIPA) buffer (1% Triton X-100, 1% sodium deoxycholate, 0.1% SDS, 50 mM TEA, pH 7.4, 150 mM NaCl).

Then, the beads were suspended in 46.5 µL resuspension buffer (4% SDS, 50 mM TEA, pH 7.4, 150 mM NaCl), followed by adding 3.5 µL freshly prepared click chemistry reaction mixture including 1 µL az-Rho (100 µM), 0.5 µL TBTA (100 µM), 1 µL TCEP (1 mM), and 1 µL CuSO_4_·5H_2_O (1 mM), with an overall reaction volume of 50 µL. The reactions were kept on for 1 h at room temperature in the dark, then 10 µL 6× SDS-loading buffer were added to terminate the reactions.

Samples were heated for 10 min at 100 °C and subjected to SDS-PAGE. One gel was loaded for Western blot analysis and the other gel was analyzed for in-gel fluorescence scanning. Proteins separated by SDS-PAGE were visualized by soaking the gel in 40% methanol, 10% acetic acid with shaking for at least 1 h and the rhodamine-associated signal was detected at excitation 532 nm/emission 580 nm using a Tanon-5200 Multi imager.

### 2.5. Cell-to-Cell Fusion Assay

HEK293T cells were transfected with the BEnv constructs together with a transactivator-expressing plasmid (BTas or Tat). At 48 h post-transfection, the transfected HEK293T cells were dissociated and the equivalent number of cells (1/10 of the total) were added at a 1:5 ratio to 1.0 × 10^5^ indicator cells (BFVL or TZM-bl) in 12-well plates. Co-culture of transfected HEK293T and indicator cells continued for 48 h, after which cells were lyzed for measuring luciferase activity.

### 2.6. Analysis of BFV SVP

BFV SVPs (including Env-only and Gag–Env SVP) were pelleted from the culture supernatants of transfected HEK293T cells by ultracentrifugation through a 20% (*w*/*v*) sucrose cushion. Briefly, equal volumes of culture supernatants (8 mL) were filtered through 0.45 µm membranes and layered onto a 20% (*w*/*v*) sucrose cushion in the PBS (1 mL) and then were ultracentrifuged at 210,000× *g* for 90 min at 4 °C (Optima LE-80K, Beckman Coulter, Fullerton, CA, USA). The pelleted BFV SVPs were resuspended in 50 µL loading buffer containing 2% SDS and analyzed by Western blotting.

### 2.7. Cell-Free Infection

HEK293T or BHK-21 cells were transfected with indicated BFV infectious clones. At 48 h post-transfection, cell-free culture supernatants were collected and filtered with a 0.45 µm membrane. Equivalent volumes of the supernatants were used to infect 2.0 × 10^5^ BFVL cells in 12-well plates. After 48 h, the infectivity of the cell-free virus was measured by luciferase activity.

### 2.8. Cell Co-Culture Assay

HEK293T or BHK-21 cells were transfected with indicated BFV infectious clones. At 48 h post-transfection, 1/20 of the total transfected HEK293T cells were harvested and co-cultured with 1.0 × 10^5^ BFVL cells in 12-well plates. At 48 h post-co-culture, luciferase activity was measured to quantitate viral replication.

### 2.9. Luciferase Reporter Assay

Cells were harvested 48 h after infection or co-culture and processed for measuring luciferase activity using a luciferase reporter assay system kit (Promega, Madison, WI, USA) following the manufacturer’s protocol.

### 2.10. Immunofluorescence Confocal Microscopy

HeLa cells were grown on 20 mm diameter glass coverslips before being assayed. The transfected cells were fixed with 4% paraformaldehyde for 10 min, washed with PBS, and permeabilized with 0.5% Triton-X 100 for 10 min. The permeabilized cells were blocked with a blocking buffer (PBS containing 5% bovine serum albumin (BSA) and 5% FBS), washed with PBS. Then, the cells were successively incubated with the primary antibody (1:500 dilution of the mouse anti-HA antibody) for 2 h and then with the secondary antibody (1:1000 dilution of the Alexa Fluor-488-conjugated goat anti-mouse antibody) for 1 h. Nuclei were stained with 0.5 µg/mL DAPI for 20 min. The coverslips were mounted onto glass slides and the images were obtained with a Leica confocal microscope.

### 2.11. Membrane–Cytosol Fractionation Assays

HEK293T cells were seeded on a 10 cm dish and transfected with 3 μg plasmid as indicated. Forty-eight hours after transfection, the cells were washed with the pre-chilled PBS and collected by centrifugation at 1000× *g* for 5 min at 4 °C. Then, the membrane and cytosol proteins were extracted using a Membrane and Cytosol Protein Extraction Kit (catalog No. P0033, Beyotime, Shanghai, China). Briefly, the collected cells were gently and fully resuspended in 800 μL membrane protein isolation solution A (added with the protease inhibitor cocktail before use) and placed on ice for 10 to 15 min. Then, the cells were homogenized for 30 to 50 times on ice until at least 80% cells were broken. The homogenates were centrifuged at 1000× *g* for 5 min at 4 °C. The supernatants were transferred into a new centrifuge tube and then centrifuged at 14,000× *g* for 30 min at 4 °C. The resulting supernatants were the cytosol fraction. The pellets were resuspended in 100 μL membrane protein isolation solution B and centrifuged at 14,000× *g* for 5 min at 4 °C. The resulting supernatants were the membrane fraction.

### 2.12. Statistical Analysis

Data were represented as means ± standard deviation (SD) from the results of three independent experiments in all bar graphs. Statistical difference between the two groups was analyzed using the Student’s t-test with GraphPad Prism version 8.0 (GraphPad software, San Diego, CA, USA). Differences were considered statistically significant when the *p*-value was < 0.05. On the figures, *p*-values are indicated as follows: * *p* < 0.05, ** *p* < 0.001, *** *p* < 0.0001, and ns for *p* > 0.05.

## 3. Results

### 3.1. Subsection

#### 3.1.1. BEnv is Palmitoylated at Amino Acid Positions C58 and C59

As membrane proteins, Env glycoproteins of various viruses have been reported to be modified and regulated by palmitoylation [20,21]. To investigate whether BEnv can also be palmitoylated, 3HA-BEnv was transfected into HEK293T cells. IFITM3 has been reported as a palmitoylated protein and was used in this study as the positive control [37]. The levels of BEnv palmitoylation were measured by click chemistry, a well-established protein palmitoylation detection approach, which is based on the reaction between alkynyl fatty acid analog alk-16 and fluorescent detection tag az-Rho. The fluorescence intensity of az-Rho indicates the protein palmitoylation level. The results showed that the az-Rho fluorescent signal was visualized only in the protein sample that was metabolically labeled with alk-16 but not in the control with dimethyl sulfoxide (DMSO), indicating that BEnv is modified by palmitoylation (Figure 1A).

BEnv has 25 cysteines. To identify the potential palmitoylation site(s), CSS-Palm version 4.0 software (http://csspalm.biocuckoo.org) was employed to predict the potential palmitoylation sites in BEnv [38]. Two palmitoylation sites C58 and C59 were predicted (Figure 1B). To validate this prediction, these two candidate cysteines were mutated individually or in combination to serine and the mutated BEnv DNA constructs were transfected into HEK293T cells. The total cell lysates were prepared and subjected to the click chemistry reaction to determine the palmitoylation levels. As shown in Figure 1C,D, mutants C58S and C59S exhibited lower BEnv palmitoylation levels and the double mutant C58/59S lost palmitoylation, akin to the effect of treatment with a general protein palmitoylation inhibitor 2-bromopalmitate (2-BP). The expression levels of different BEnv mutants were similar to the wild type BEnv. These results identify amino acids C58 and C59 as the two palmitoylation sites in BEnv.

#### 3.1.2. DHHC3, DHHC7, and DHHC20 Palmitoylate BEnv in Cells

To identify host DHHC proteins involved in BEnv palmitoylation, we conducted a co-expression screening assay. Each of the 23 mammalian DHHCs together was transfected into HEK293T cells together with BEnv to determine which DHHCs modify BEnv. The results showed that the levels of BEnv palmitoylation was increased to different degrees as a result of overexpression of different DHHC proteins, with the greatest increase by DHHC3, followed by DHHC7 and DHHC20 (Figure 1E–I). Therefore, DHHC3, DHHC7, and DHHC20 act as PATs for BEnv.

#### 3.1.3. BDHHC3 and BDHHC20 Interact with BEnv and Palmitoylate BEnv at both C58 and C59 Residues

Since the DHHCs used in the above screening assays are from humans, not of bovine origin, and human and bovine DHHCs share 80% similarity in the amino acid sequences, bovine DHHCs (BDDHCs) were thus further tested to confirm the above results. Two BDHHCs were constructed, including BDHHC3 and BDHHC20. BDHHC7 was not studied partly because of the difficulty of cloning its cDNA into the expression vector. In addition, BDHHC7 shares 77% similarity in the amino acid sequence with BDHHC3. Both enzymes exhibit the same membrane topology, the same tissue and cell type-specific expression patterns and are substantially overlapped in their substrates [27,29]. We expect that results with BDHHC3 can inform the potential effect of BDHHC7 on BFV Env. The results showed that wild type BDHHC3 and BDHHC20 increased the palmitoylation level of BEnv, while mutants BDHHC3-C157S and BDHHC20-C156S carrying mutations in the DHHC catalytic motif did not palmitoylate BEnv (Figure 2A,B).

Palmitoylation requires the binding of a DHHC protein with its substrates. To examine whether BDHHC3 and BDHHC20 interact with BEnv, cells were co-transfected with 3HA-BEnv and an empty vector or Flag-tagged DHHCs. Immunoprecipitation of both Flag-BDHHC3 and Flag-BDHHC20 from total cell lysates with the anti-Flag antibody resulted in co-precipitation of 3HA-BEnv, not in controls without Flag-BDHHC expression. We also observed that the level of BEnv co-precipitated by Flag-BDHHC3 was significantly higher than that by Flag-BDHHC20 (Figure 2C, top panel). Therefore, BEnv interacts with BDHHC3 and BDHHC20, with stronger binding to BDHHC3.

Since BDHHC3 and BDHHC20 have different degrees of interaction with BEnv, this may affect catalytic efficiency. To compare the efficacy with which BDHHC3 and BDHHC20 palmitoylate BEnv, we next measured the levels of BEnv palmitoylation in response to different doses of co-transfected BDHHC3 and BDHHC20. As shown in Figure 2D,E, the lowest amount of BDHHC3 or BDHHC20, 50 ng BDHHC3 and BDHHC20, already drastically enhanced BEnv palmitoylation, and the level of BEnv palmitoylation did not increase further with more than 200 ng BDHHC3 and BDHHC20 used in transfection. These results suggest that although BDHHC20 interacts with BEnv relatively weakly, BDHHC20 can efficiently palmitoylate BEnv.

Next, to determine which of the two palmitoylation sites, C58 and C59, of BEnv is modified by BDHHC3 and BDHHC20, we generated the single or double palmitoylation site mutants of BEnv and expressed them along with BDHHC3 or BDHHC20. We found that both BDHHCs were capable of palmitoylating both C58 and C59 of BEnv (Figure 2F,G).

Taken together, these results demonstrate that BDHHC3 and BDHHC20 interact with BEnv and palmitoylate at C58 and C59.

#### 3.1.4. Palmitoylation of BEnv Is Essential for Its Membrane Fusion Activity

Viral membrane glycoproteins are involved in virus entry through binding to receptor and promoting membrane fusion [13]. To determine the potential effect of palmitoylation on a BEnv-mediated membrane, we conducted an Env-mediated membrane fusion assay. HEK293T cells were co-transfected with the wild type or palmitoylation site-mutated BEnv and BTas. After 48 h, the transfected cells were co-cultured with the BFVL indicator cells. BEnv-mediated membrane fusion of donor cells HEK293T and acceptors cells BFVL allowed BTas to activate the expression of the luciferase reporter gene in BFVL cells (Figure 3A). The results showed that the membrane fusion ability was significantly impaired in BEnv C58S and was almost abrogated in BEnv C58/59S, while BEnv C59S still allowed substantial fusion activity and retained 70% of fusion capacity compared to wild type BEnv (Figure 3B). Similar results were observed by using Tat and TZM-bl indicator cells, in which BEnv-mediated cell–cell membrane fusion allows Tat to activate expression of the luciferase reporter gene in TZM-bl cells (Figure 3C). Taken together, these results demonstrate that palmitoylation of BEnv at C58 is indispensable for its fusion activity.

#### 3.1.5. Palmitoylation of BEnv Is Critical for BFV SVP Release

Env is essential for the budding and release of FV particles [14]. We thus examined whether BEnv palmitoylation is required for BFV SVP release. Results of the Western blot analysis showed that different BEnv mutants were expressed at the same level as the wild type BEnv (Figure 4). Strikingly, the BEnv C58S and BEnv C58/59S mutants dramatically decreased or even abolished the SVP production mediated by BEnv alone or together with BEnv–BGag, while BEnv C59S exerted no effect (Figure 4). Therefore, palmitoylation of C58 is essential for the function of BEnv in supporting the production of SVP.

#### 3.1.6. BEnv Palmitoylation Is Required for BFV Replication

Since mutation of palmitoylation sites in BEnv significantly impaired its fusion ability and particle release, we posited that viral replication may depend on BEnv palmitoylation. To determine whether BEnv palmitoylation is required for BFV replication, we generated BFV infectious clones with mutated BEnv palmitoylation sites, including pcBFV-Env C58S, pcBFV-Env C59S, and pcBFV-Env C58/59S. At 48 h post-transfection, the culture supernatants (cell-free BFV particles) and 1/20 of the total transfected HEK293T cells (cell-associated BFV particles) were incubated with the BFVL indicator cell line; the rest of the cells were collected for Western blotting analysis. The results showed that levels of cell-free and cell-associated BFV were reduced significantly for both pcBFV-Env C58S and pcBFV-Env C58/59S, whereas pcBFV-Env C59S showed a moderate decrease (Figure 5A). Consistent with this observation, the expression levels of BFV Gag, Env, and BTas were also reduced in pcBFV-Env C58S and pcBFV-Env C58/59S in the transfected cells compared to those in control cells (Figure 5A). To confirm the above results, the same experiments were carried out in BHK-21 cells; similar results were observed (Figure 5B). Collectively, these data suggest that palmitoylation of BEnv, especially at C58, is important for BFV replication.

#### 3.1.7. Palmitoylation of BEnv Is Required for Its Trafficking to the Cell Surface

Palmitoylation has been reported to target proteins to the lipid rafts [19]. We thus examined whether trafficking and subcellular localization of BEnv is dependent on palmitoylation by performing immunofluorescence assays. As shown in Figure 6A, the cellular surface expression levels of the two mutants, BEnv C58S and BEnv C58/59S, were dramatically reduced. Similar observation was made when 2-BP was used to prevent BEnv palmitoylation. In contrast, the mutant BEnv C59S displayed a phenotype similar to the wild type BEnv. We further examined the subcellular localization of wild type BEnv and its mutants (C58S, C59S, and C58/59S) by performing a membrane fractionation assay. A large portion of wild type BEnv and BEnv C59S was observed in the membrane fraction, as opposed to the two mutants, BEnv C58S and BEnv C58/59S, that were mostly detected in the cytosol fraction (Figure 6B). These results suggest that palmitoylation regulates BEnv trafficking to and stable association with the cell surface.

## 4. Discussion

The host cellular palmitoylation machinery is hijacked by viruses to establish the structure and function of viral proteins, ensure efficient viral replication [20,39]. In this study, we report that palmitoylation of BEnv regulates BEnv targeting to the cell surface, is indispensable for the membrane fusion activity of BEnv and its budding and incorporation into SVP, and thus warrants efficient replication of BFV. These results further highlight the important role that palmitoylation plays in Env function and FV replication, provide the first evidence that FV hijacks host DHHCs to modify and regulate its Env protein.

Among the viral transmembrane proteins that have been reported to undergo palmitoylation, the modified cysteine residues are mostly located in the cytoplasmic tail, often close to the hydrophobic transmembrane domain (TMD) [20,21,30]. For example, in the G protein of the vesicular stomatitis virus (VSV), hemagglutinin (HA) of the influenza A virus (IAV), S and E proteins of the severe acute respiratory syndrome coronavirus (SARS-CoV), the palmitoylated cysteines are all located in cytoplasmic tails, 0 to 9 amino acids away from the TMD [40,41,42,43,44]. In this study, we identified C58 and C59 as the two palmitoylation sites in BEnv. It is noted that C58 and C59 are also located at the N-terminal cytoplasmic tail of the LP subunit, and 9 amino acid residues away from the hydrophobic TMD, which is consistent with the positioning of palmitoylation sites of Env from other viruses. The other 23 cysteines in BEnv are not acylated, probably because they are either too far away from the TMD or embedded in the membrane bilayer which cannot be accessed by the DHHC proteins. Importantly, these two palmitoylation sites in BEnv are evolutionarily conserved among most different primate FV strains [12,45], which further support the important functions of palmitoylation of these two cysteines in the function of FV Env proteins.

By screening for cellular DHHCs that target BEnv, we found three human DHHC proteins, DHHC3, DHHC7, and DHHC20, that may serve as candidate PATs of BEnv. Bovine BDHHC3 and BDHHC20 are highly homologous with human DHHC3 and DHHC20, with 97.32% and 88.22% similarity in their amino acid sequences. It is thus not surprising that bovine BDHHC3 and BDHHC20 also palmitoylate BEnv. Michael Veit and Mathieu Blanc reported that acylation of viral proteins often occurs along the secretory pathway at the ER and/or Golgi apparatus where most mammalian DHHC enzymes are located [20,46]. Given that BEnv is synthesized at the ER and secreted though the Golgi apparatus [16], it is assuring to observe that BEnv are palmitoylated by the Golgi-localized DHHC3, DHHC7, DHHC20, BDHHC3, and BDHHC20. The redundancy of DHHCs in mediating BEnv palmitoylation may be one mechanism to ensure palmitoylation and function of BEnv, thus warranting efficient replication of BFV. Moreover, studies have shown that some host cellular proteins are also palmitoylated by more than one DHHC. For instance, more than half of the DHHCs increased the palmitoylation level of IFITM3, with DHHC 3, 7, 15, and 20 showing the greatest effect [37]. DHHC3, DHHC7, and DHHC17 cause SNAP25 palmitoylation [47]. In regard to viral proteins, chikungunya virus NSP1 is palmitoylated by DHHC2 and DHHC19, HSV-1 UL20 and HIV-1 Tat are palmitoylated by DHHC3 and DHHC20 [31,32,33,34].

The PAT activity of DHHCs appears to be quite potent, since a very little amount of BDHHC3 and BDHHC20 can palmitoylate BEnv to very significant degrees. This may explain why the expression level of DH’HCs did not affect our screening results shown in Figure 1. In addition, although our results showed BDHHC3 and BDHHC20 catalyze BEnv palmitoylation at both C58 and C59, we are uncertain whether the individual BDHHC has any preference for C58 or C59.

Palmitoylation of virus Env proteins is often crucial for their functions and hence for viral replication [21]. Palmitoylation of the HIV-1 Env is essential for Env incorporation into virions and maintaining viral infectivity [22]. Palmitoylation of IAV HA is required for membrane fusion, incorporation into virions, and assembly of virus-like particles (VLP) [41,42]. Palmitoylation of M-MuLV Env determines lipid raft binding and cell surface localization [23]. Palmitoylation of the SARS-CoV S protein is essential for its fusion activity and virion assembly [43]. Fusion activity of the FIV Env and its incorporation into virions are also dependent on its proper palmitoylation [24]. We now report that BEnv palmitoylation is necessary for its plasma membrane localization, membrane fusion activity, SVP release, and BFV replication. Palmitoylation provides a hydrophobic anchor, enhances membrane affinity of the modified proteins, thus enables protein trafficking to and interacting with cellular membranes [19]. For enveloped viruses, the number of viral envelope proteins in the virion is closely related to their fusion activity [21]. It has been shown that palmitoylation of IAV HA increases its plasma membrane expression, thereby improving virus particle budding and membrane fusogenicity [42]. Our data also showed that palmitoylation promotes BEnv transportation to the plasma membrane and enriches BEnv on the cell surface, which enables virion formation and subsequent fusion of virions with target cells. One exception is that although VSV G protein is also palmitoylated, this modification is not necessary for membrane fusion nor for incorporation into virions [40]. It is possible that the VSV G protein may use other mechanisms to facilitate its viral functions.

We observed that both C58S and C58S/C59S mutations drastically decrease cell surface expression of BEnv, its cell–cell membrane fusion activity, and SVP budding. However, C59S alone had only a moderate effect on the function of BEnv and BFV replication. These results suggest that palmitoylation of C58 is more critical for the function of BEnv and BFV replication compared to palmitoylation of C59 which cannot compensate for the loss of palmitoylation on the C58. Similar observations have also been made with RSV Env, which also have two palmitoylation sites at C164 and C167. In contrast to the C167G mutation which has little effect on the RSV Env function, mutations C164G and C164G/C167G exert a severe deleterious effect [25]. One possibility is that palmitoylation of only one specific cysteine, C58 in BEnv or C164 in RSV Env, can support the protein function, which cannot be substituted by palmitoylation at a different site.

In summary, we report for the first time that BEnv undergoes palmitoylation at amino acid residues C58 and C59 by BDHHC3 and BDHHC20. This modification allows BEnv trafficking to the cell surface, increases BEnv membrane fusion ability and BEnv-mediated SVP release, and thus ensures efficient BFV replication. Hence, our findings provide new insights into the role of post-translational modifications in the function of FV Env, may advance the development of FV vectors, and also suggest host DHHCs as potential targets of antiviral drugs.

## Figures and Tables

**Figure 1 viruses-13-00031-f001:**
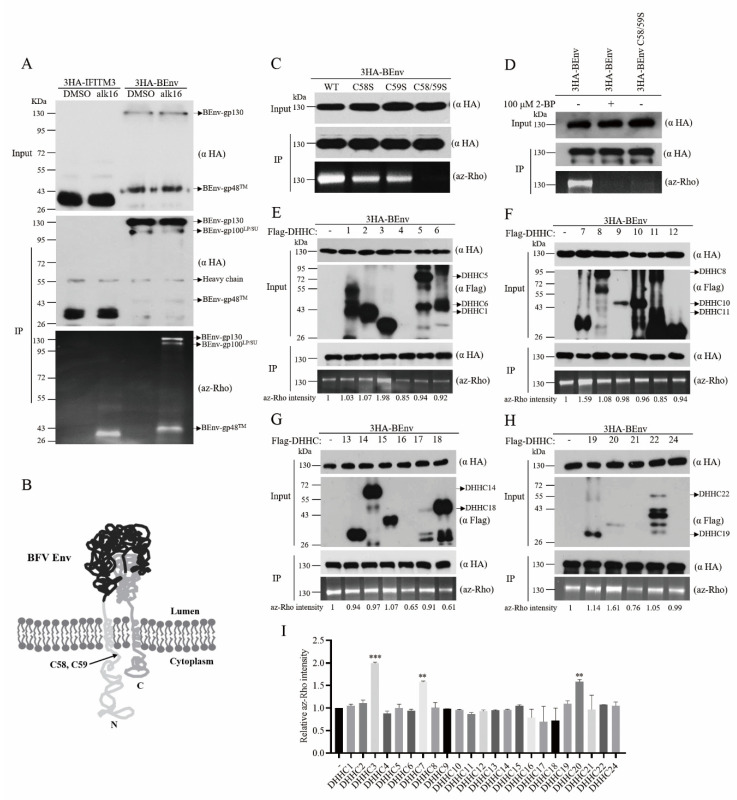
BEnv is palmitoylated at amino acid positions C58 and C59 by DHHCs. (**A**) HEK293T cells were transfected with 3HA-BEnv or 3HA-IFITM3 as the positive control of protein palmitoylation. Cells were treated with alk-16 or DMSO (negative control) for 2 h. The immunoprecipitated BEnv or IFITM3 were subjected to reaction with az-Rho via click chemistry. Levels of protein palmitoylation were determined by in-gel fluorescence scanning. (**B**) Membrane topology of the BFV Env. The palmitoylation sites C58 and C59 predicted by CSS-Palm version 4.0 are indicated. (**C**,**D**) Wild type BEnv or cysteine mutants (C58S, C59S, and C58/59S) lacking specific palmitoylation sites were transfected into HEK293T cells. Cells were labeled with alk-16 in the presence of 2-BP and lyzed prior to immunoprecipitation of BEnv and reaction with az-Rho via click chemistry for visualization of palmitoylation by fluorescence gel scanning. Western blotting was performed to confirm the comparable expression of both WT and mutant BEnv. (**E**–**H**) HEK293T cells were co-transfected with 3HA-BEnv and either control vector or different Flag-DHHCs. Cells were then metabolically labeled with alk-16. BEnv palmitoylation was examined by click chemistry and in-gel fluorescence scanning. Western blots show the expression of BEnv and DHHCs. The az-Rho (palmitoylation signal) intensities were quantified with Image J. The above numbers represent different DHHCs. (**I**) The palmitoylation signal intensities from three independent repeated experiments are shown in the bar graph. *p*-values are indicated as follows: ** *p* < 0.001, *** *p* < 0.0001.

**Figure 2 viruses-13-00031-f002:**
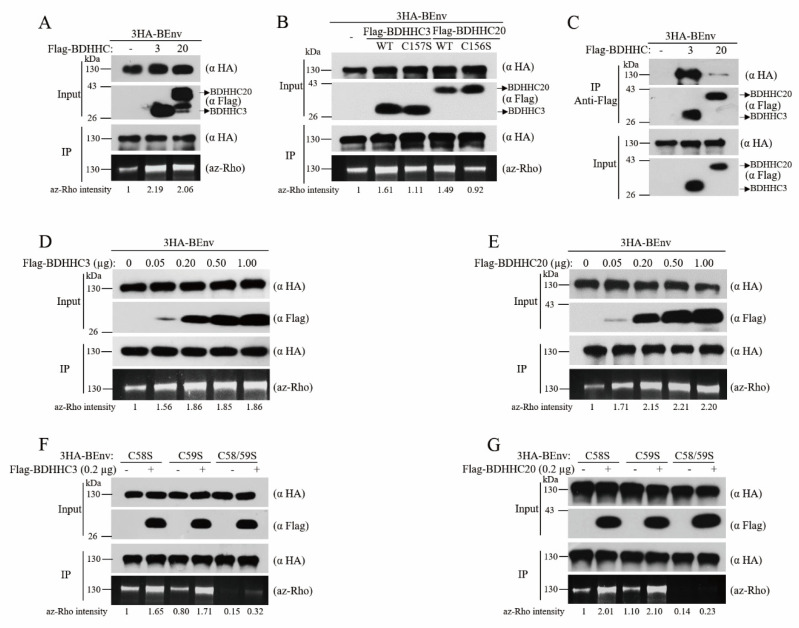
BDHHC3 and BDHHC20 interact with BEnv and palmitoylate BEnv at C58 and C59. (**A**,**B**) HEK293T cells were co-transfected with BEnv and either wild type BDHHC3, BDHHC20, or their mutants (BDHHC3-C157S and BDHHC20-C156S). Levels of BEnv palmitoylation were evaluated as described in Figure 1. The az-Rho intensities were quantified with Image J. (**C**) HEK293T cells were co-transfected with 3HA-BEnv and either Flag-BDHHC3, Flag-BDHHC20, or an empty vector. Cell lysates were subjected to immunoprecipitation with the anti-Flag antibody. The co-immunoprecipitated BEnv was analyzed using the anti-HA antibody (the top panel). Ten percent input of total extracts was probed with anti-HA and anti-Flag to verify expression of 3HA-BEnv, Flag-BDHHC3, Flag-BDHHC20. (**D**,**E**) 3HA-BEnv was co-transfected into HEK293T cells with increasing doses of expression plasmids coding for either Flag-BDHHC3 or Flag-BDHHC20. Then, the total cell lysates were prepared and subjected to the click chemistry reaction to evaluate the levels of BEnv palmitoylation. The az-Rho intensities were quantified with Image J. (**F**,**G**) HEK293T cells were co-transfected with BEnv palmitoylation site mutants (C58S, C59S, and C58/59S) and either BDHHC3 or BDHHC20. Levels of BEnv palmitoylation were visualized by az-Rho fluorescence gel scanning. The az-Rho intensities were quantified with Image J.

**Figure 3 viruses-13-00031-f003:**
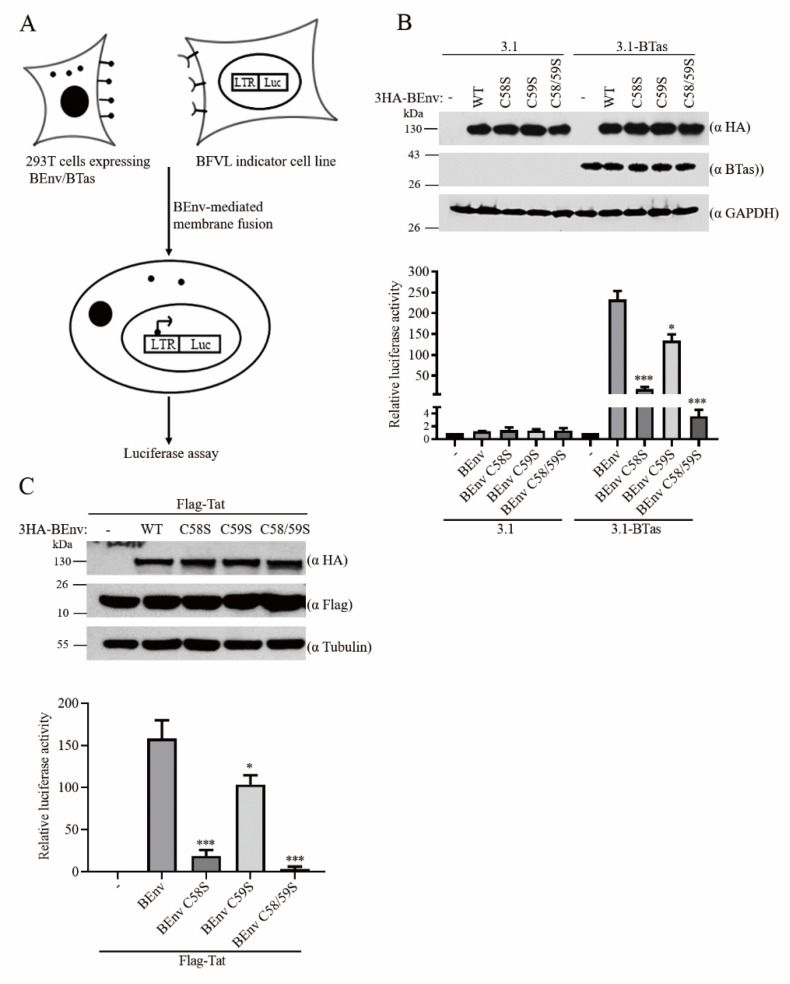
BEnv palmitoylation is essential for its membrane fusion activity. (**A**) Schematic of the BEnv-mediated membrane fusion assay. (**B**,**C**) HEK293T cells were transfected with wild type or mutant BEnv DNAs together with BTas or Tat DNAs as indicated. Forty-eight hours post-transfection, the transfected HEK293T cells (1/10 of the total) were co-cultured with BFVL or TZM-bl indicator cell lines and the luciferase activity was measured. A fraction of the transfected cells was lyzed and subjected to immunoblotting to monitor BEnv and BTas or Tat expression. Data presented were obtained from three independent experiments. *p*-values are indicated as follows: * *p* < 0.05, *** *p* < 0.0001. GAPDH, glyceraldehyde-3phosphate dehydrogenase.

**Figure 4 viruses-13-00031-f004:**
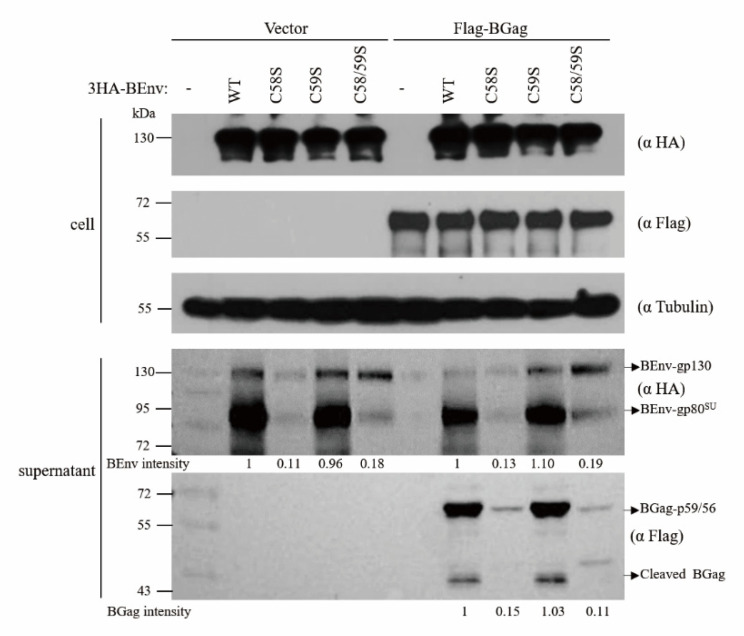
BEnv palmitoylation is critical for BFV SVP release. HEK293T cells were transfected with either wild type or mutant 3HA-BEnv and Flag-BGag as indicated. After 48 h, cells were lyzed and subviral particles were harvested by ultracentrifugation through a 20% sucrose (*w*/*v*) cushion from cell culture supernatants as described in the Materials and methods. BEnv and BGag in cell lysates and viral particles were detected by Western blotting using the indicated antibodies. The indicated band intensities were quantified with Image J.

**Figure 5 viruses-13-00031-f005:**
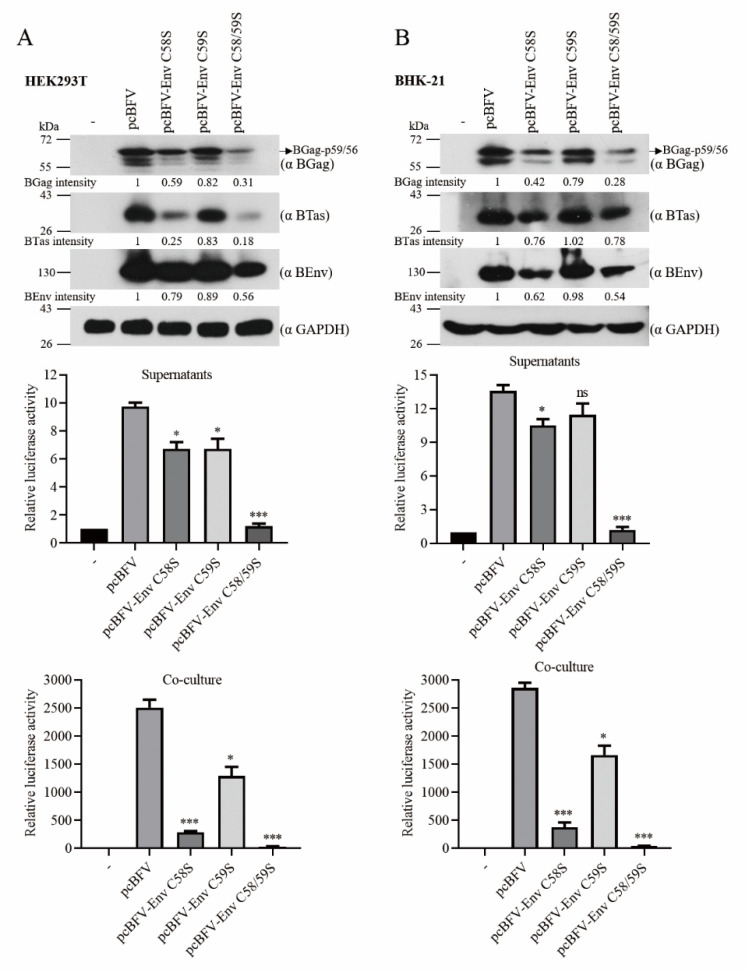
BEnv palmitoylation is required for BFV transmission. (**A**) HEK293T cells were transfected with an empty vector, wild type BFV infectious DNA clone pcBFV, or BFV infectious DNA clone with mutations in BEnv-palmitoylated cysteines (pcBFV-Env C58S, pcBFV-Env C59S, pcBFV-Env C58/59S). At 48 h post-transfection, culture supernatants or transfected 293T cells (1/20 of the total) were co-cultured with the BFVL indicator cell line for another 48 h and luciferase activity was measured (right). The data shown were the averages from three independent experiments; error bars represent the SD. *p*-values are indicated as follows: * *p* < 0.05, *** *p* < 0.0001, and ns for *p* > 0.05. The rest of the transfected cells were subjected to Western blotting with the indicated antibodies (left). The indicated band intensities were quantified with Image J. (**B**) Same as in A, except that it was performed in BHK-21 cells.

**Figure 6 viruses-13-00031-f006:**
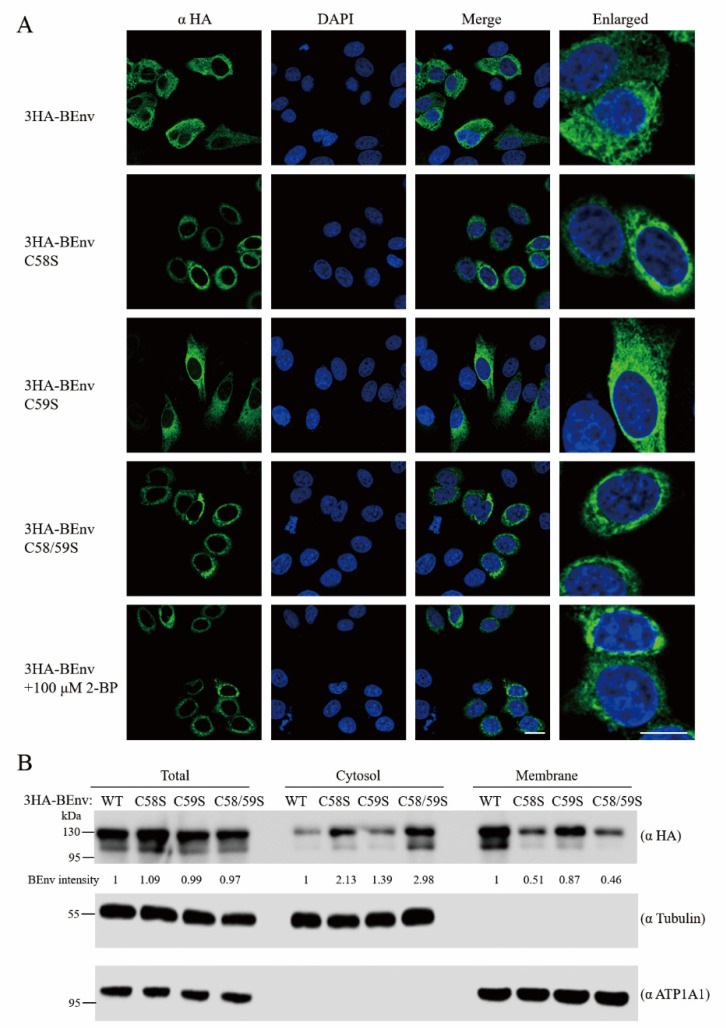
(**A**) Palmitoylation is required for trafficking of BEnv to the cell surface. HeLa cells were transfected with either wild type or palmitoylation mutant 3HA-BEnv. Cells transfected with wild type BEnv were treated with 100 µM 2-BP for 6 h before being assayed. Twenty-four hours post-transfection, the cells were subjected to immunofluorescence for detection of BEnv using the mouse anti-HA antibody. The nuclei were stained with DAPI. The subcellular localization of BEnv was analyzed by confocal laser scanning microscopy. Scale bars represent 10 μm. (**B**) HEK293T cells were transfected with either wild type 3HA-BEnv or its mutants (C58S, C59S, and C58/59S). After 48 h, cells were collected, the membrane and cytosol fractions were prepared as described in the Materials and methods. Each fraction was examined by Western blotting to measure the levels of BEnv. Tubulin and ATP1A1 were probed as markers for the cytosol and plasma membrane, respectively. Intensity of protein bands was determined with Image J.

## Data Availability

The data presented in this study are available on request from the corresponding author.

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
