# Peer review of "Palmitoylation of the Bovine Foamy Virus Envelope Glycoprotein Is Required for Viral Replication"

_viruses, 2020, doi:10.3390/v13010031_

Round 1

Reviewer 1 Report

This study described the post translational modification of Env from bovine Faomy virus (BFV) and its role in viral replication. The author clearly showed that Env is palmytoylated and identified both the amino acid that supported this modification and the cellular enzyme responsible for this modification. However, the role of Env modification in viral replication is less convincing, in part because some control are missing or because figure could be reorganized. Specifically, a decrease of Env surface expression could explain the lower infectivity of the palmytoylation mutants as well as the lower incorporation of Env mutant in viral particles. This should be controlled first and before the analysis of viral transmission.

Specifically here are my comments :

Fig 1 : it woud be more convincing if the author could show quantification of palmytoylation signal in each panel. Please also indicate in each panel (and each figures) the molecular weight and the size of the expected band with an arrow for example. Could you please comment what is the lower band detected in the alk16 /BEnv lane ?

Please specify in the legend that the numbers in panels E; F G; H are related to the different DHHC. One could think that they indicate lane numbers.

Lane 233 : the sentence is not clear, it seems that some arguments are missing. Please correct. Authors should also mention the level of conservation between Human and bovine DHHC.

FiG 2 : add molecular weight and quantification of palmytoylation signal.

Fig3 : What is the level of BFV viral particle produced with the different ENV ? What is the infectivity after normalization to the number of viral particle ?

The level of TAS is reduced in the 293T transfected cells could the author comment, since Env is not supposed to control Tas expression ?

The Luciferase activity could rely on Tas transferred to the target reporter cells without productive infection, specially because the level Tas expression is decreased after transfection of Env mutants. Could the author adjust the level of Tas expression or control that no Tas is transferred without productive infection of target cells ?

The membrane expression of Env and its mutant should be controlled to avoid a trivial decrease in infectivity due to lower Env surface expression. Similarly Env incorporation in virion released from the transfected cells should be controlled.

This figure would be more convincing if placed after fig 4 and 5 because it confirm that Env mutation induced a decrease in both particles release and fusion ability.

Lane 289 : I suggest to replace viral replication by viral transmission that is more accurate to the experiment shown.

Fig 4 the author should control Env wt and mutant surface expression and express results relative to this surface expression.

Fig 5 Please indicate the molecular weight. Please indicate the expected size of env and gag proteins since several bands are detected in the gel from supernatants.

Please quantify the level of each Env and Gag protein relative to WT.

Fig 6 the surface expression is not visible in this figure since  all env wt and mutants appear to be in the cytoplasm. Please include a surface marker to show the colocalization and provided higher magnification to fully appreciate the subcelluar localization. In addition, subcellular localization analyzed by IF should be controlled by sub cellular fractionation and WB on the different fractions with adequate controls.

Reviewer 2 Report

This manuscript describes experiments demonstrating that the bovine foamy virus envelope protein is palmitoylated at two specific residues by several host PATs and that palmitoylation is necessary for Env trafficking and infectivity. The study is generally thorough and well-described. However, there are several problems with study design and analysis of the data that should be addressed so that all the conclusions are justified by the results.

Primary comments:

  1. All of the palmitoylation experiments are conducted in human rather than bovine cells. The results would be substantially more convincing with evidence that the bovine foamy virus envelope is palmitoylated in bovine cells.
  2. The immunoblots depicted in the manuscript have no markers of protein size. It would be helpful to have protein size markers added to the plots. For instance, do the bands in Fig. 1C correspond to the larger or smaller molecular weight bands shown in Fig 1A?
  3. In Fig 1A, what do the low and high molecular weight bands for 3HA-BEnv correspond to? Is the low MW band a truncated or cleaved version of Env?
  4. The results depicted in Fig 1 E-H do not appear quite so clear cut as the manuscript describes. Levels of palmitoylation for DHHCs 3, 7, and 20 are not dramatically different from other DHHCs. How repeatable and statistically significant is this? It would be more convincing if the fluorescence were quantified for each DHHC compared to control, performed with replicates, displayed on a graph, and analyzed statistically to determine whether the increases for 3, 7, and 20 are significant (and whether there are significant increases for any of the other DHHCs).
  5. Typically in retroviral infectivity assays, the viruses produced following transfection are quantified, then virus input is standardized for infection. This would allow determination of whether the mutant viruses are less infectious than wild-type or whether the mutations result in less virus production following transfection or both.
  6. I do not find the results depicted in Fig. 5 convincing. Why do the Env vector only lanes (no Gag) have strong anti-HA (Env) detection in the viral particles prep? There should be no viral particles without Gag. Why use “subviral particles” at all? Why not use the full length molecular clones to produce virus and then purify these viruses?

Minor comments:

  1. Line 235-236. Explanation for why BDHHC7 was not cloned is weak – 77% identity is not enough to assume identical function.
  2. Introduction should briefly describe the entry receptor for BFV – is it known?
  3. Line 27, word missing, “that cause endemic infection in non-human”
  4. Section 2.1 methods. State whether the HA and FLAG tags are N or C-terminal (presumably C terminal). Which restriction enzymes were used for gene insertion?
  5. Line 98. Provide more information on the cell lines. What are BFVL cells? What are TZM-bl cells?
  6. Line 233-234 grammar. First sentence is not a sentence, rephrase.
  7. Line 237 spelling. “wide-type” should be “wild-type”
  8. Line 242. “Flag-BDHHC7” should be “Flag-BDHHC20”

Round 2

Reviewer 2 Report

The authors have responded to all comments and provided important additional information in the manuscript. Furthermore, additional analyses and experiments have been conducted that improve the quality of the results and clarify the conclusions. In particular, clarity of the figures has been improved with appropriate labelling. I do not have further concerns regarding the scientific content or conclusions.

The quality of the text is overall good, but could use a thorough edit to correct sporadic grammatical errors.